# Effects of N-acetylcysteine on Growth, Viability, and Ultrastructure of In Vitro Cultured Bovine Secondary Follicles

**DOI:** 10.3390/ani12223190

**Published:** 2022-11-17

**Authors:** Danisvânia R. Nascimento, Venância A. N. Azevedo, Pedro A. A. Barroso, Laryssa G. Barrozo, Bianca R. Silva, Anderson W. B. Silva, Mariana A. M. Donato, Christina A. Peixoto, José R. V. Silva

**Affiliations:** 1Laboratory of Biotechnology and Physiology of Reproduction (LABIREP), Federal University of Ceara, Av. Comandante Maurocélio Rocha Ponte 100, Sobral CEP 62041-040, CE, Brazil; 2Laboratory of Ultrastructure, CPqAM/FIOCRUZ, Federal University of Pernambuco, Recife CEP 50670-901, PE, Brazil

**Keywords:** viability, growth, NAC, secondary follicle

## Abstract

**Simple Summary:**

During in vitro culturing of secondary follicles, several factors, such as exposure to light, large volumes of culture media, and temperature variation, can promote oxidative stress, which can cause damage to oocyte and granulosa cells. To minimize the adverse effects of oxidative stress, this study aimed to investigate the effects of different concentrations of N-acetylcysteine (NAC) on the growth, antrum formation, viability, and ultrastructure of bovine secondary follicles cultured in vitro. The results showed that supplementation of culture medium with 1 mM NAC increased calcein staining and the growth rate in bovine secondary follicles cultured in vitro. Ultrastructural analysis showed that oocytes from follicles cultured in the control medium alone, or with 1 mM NAC, had intact zonae pellucidae in close association with oolemmae, but the ooplasm showed mitochondria with a reduced number of cristae. The presence of higher concentrations (5 or 25 mM NAC) caused damage to cell membranes and organelles, as well as reducing the percentages of growing follicles.

**Abstract:**

This study aimed to investigate the effects of different concentrations of N-acetylcysteine (NAC) on the growth, antrum formation, viability, and ultrastructure of bovine secondary follicles cultured in vitro for 18 days. To this end, the follicles were cultured in TCM-199^+^ medium alone or supplemented with 1.0, 5.0, or 25.0 mM NAC. Follicular growth, antrum formation, viability (calcein-AM and ethidium homodimer-1) and ultrastructure were evaluated at the end of culture period. The results showed that 1.0 mM NAC increased the percentage of growing follicles and the fluorescence intensity for calcein-AM when compared to other treatments (*p* < 0.05). On the other hand, follicles cultured with 25.0 mM NAC had higher fluorescence intensity for ethidium homodimer-1, which is a sign of degeneration. Ultrastructural analysis showed that oocytes from follicles cultured in control medium alone or with 1 mM NAC had intact zonae pellucidae in close association with oolemmae, but the ooplasm showed mitochondria with a reduced number of cristae. On the other hand, oocytes from follicles cultured with 5 or 25 mM NAC had extremely vacuolated cytoplasm and no recognizable organelles. In conclusion, 1 mM NAC increases cytoplasmic calcein staining and the growth rate in bovine secondary follicles cultured in vitro, but the presence of 5 or 25 mM NAC causes damage in cellular membranes and organelles, as well as reducing the percentages of growing follicles.

## 1. Introduction

High numbers of immature oocytes enclosed in early follicles are present in mammalian ovaries, and the development of in vitro culture systems to promote their growth in vitro has a great potential to increase oocyte availability for in vitro fertilization [1,2,3]. The in vitro culture of follicles at their initial stages of development, such as secondary follicles, is also important to understanding the mechanisms that control follicular growth and maturation [2,3,4,5]. The complex pathways regulating follicular growth are, however, obstacles to having competent oocytes in vitro [6]. The optimization of in vitro conditions to mimic the ovarian environment is the main challenge for researchers in the field. It is important to highlight that in-vitro-cultured follicles are not in their natural environment and need proper conditions to survive and to develop. In cows, various reports have demonstrated that secondary follicles cultured for a period of 18 days grow up to early antral follicles, but an increase in the atresia rate is commonly reported at this stage [5,7,8].

During in vitro culturing of secondary follicles, various factors, such as variation in medium osmolality, exposure to light, large volumes of culture media, variation in temperature, and static nature of culture systems, can favor oxidative stress [9,10]. A high concentration of oxygen in vitro (up to 20%) can also result in the accumulation of reactive oxygen species [11]. Oxidative stress can cause damage to the DNA, proteins, and mitochondria of oocyte and granulosa cells, as well as increase apoptosis [11,12]. It has been reported that long-term moderate oxidative stress causes ultrastructural changes in ovarian follicles and reduces fertility in mice [13]. These authors showed that the oocytes had swollen Golgi apparatus, a large number of lipid droplets, and mitochondria with vacuolization and degeneration of the cristae and matrix. To minimize the adverse effects of oxidative stress, various studies have reported the importance of adding antioxidant substances to in vitro culture media [5,8,9]. N-acetylcysteine (NAC) is a substance that can be used to optimize in vitro conditions during follicle growth [14]. The antioxidant effects of NAC occur directly by donating electrons to reactive species, and indirectly by being a precursor of glutathione that integrates the intracellular antioxidant system [15]. Several studies have emphasized the potential of NAC to improve the quality of oocytes, granulosa cells, ovarian tissues, and embryos during in vitro culturing [16,17,18,19]. A synergic action between NAC and FSH to promote preantral follicle growth and viability in cultured human ovarian tissues has been reported [20]. However, it is not known whether NAC improves growth and viability, or whether it helps to preserve the ultrastructure of bovine secondary follicles cultured in vitro.

This study aimed to investigate the effects of different concentrations of NAC on the growth, antrum formation, viability, and ultrastructure of bovine secondary follicles cultured in vitro.

## 2. Material and Methods

### 2.1. Ovarian Collection

All chemicals were purchased from Sigma Chemicals Company, St. Louis, MO, USA, unless otherwise stated in the text.

Ovaries (*n* = 60) from mixed breed cows (*Bos taurus*), aged between 4 and 12 years, were collected from a local abattoir (Sobral, Ceará, Brazil). Immediately after death, the ovaries containing visible antral follicles were washed in 70% ethanol and then washed in HEPES-buffered TCM-199 supplemented with streptomycin (0.1 mg/mL) and penicillin (100 IU). Subsequently, the ovaries were transported within 1 h to the laboratory in TCM-199 at 4 °C. This study was approved by the Ethics and Animal Welfare Committee of the Federal University of Ceará (Number 02/21).

### 2.2. Isolation and Culture of Secondary Follicles

For follicular isolation, the method of Vasconcelos et al. [8] was followed. Fragments (1–2 mm) of the ovarian cortex were removed with the aid of a sterile scalpel blade and placed in TCM-199 medium supplemented with HEPES (0.05 mM/mL) and penicillin (100 IU)/streptomycin (0.1 mg/mL). Subsequently, the fragments were analyzed in a stereomicroscope (SMZ 645 Nikon, Tokyo, Japan) to visualize the secondary follicles (150–250 μm). Follicles that did not have antral cavity were manually isolated using 25 G needles. After isolation, follicles with intact basement membranes, spherical oocytes, surrounded by granulosa cells, and intact theca cell layers were selected for culturing.

Secondary follicles were cultured individually in 100 μL microdrops of TCM-199 medium (pH 7.2–7.4) supplemented with FSH (100 ng/mL), ITS (insulin (10 μg/mL), transferrin (5.5 μg/mL), selenium (5 ng/mL), ascorbic acid (50 μg/mL), BSA (3.0 mg/mL), glutamine (2 mM), hypoxanthine (2 mM), penicillin (100 IU), streptomycin (0.1 mg/mL), and HEPES (0.05 mM) (TCM-199^+^). The follicles were randomly cultured in TCM-199^+^ alone or supplemented with 1.0, 5.0, 25.0 mM NAC. The concentrations of NAC were based on previous studies [18,19]. In vitro culturing was carried out in an incubator with 5% CO_2_ in the air, at 38.5 °C, for 18 days. Every 2 days, 60 μL of the culture medium was replaced. At the end of culturing, follicular diameters and antrum formation were analyzed under a stereomicroscope (SMZ 645 Nikon, Tokyo, Japan). This experiment was repeated eight times, and a total of 398 follicles were isolated and cultured.

### 2.3. Evaluation of Secondary Follicle Morphology, Growth, and Antrum Formation

The follicles were analyzed at days 0 and 18 of culturing using an inverted microscope (Nikon, Eclipse, TS 100, Yokohama, Japan), and the images were captured by NIS elements software (Nikon Instruments Inc., Yokohama, Japan). Follicle diameters were calculated as the average of two perpendicular measurements of the outer layer of the thecal cells solely in morphologically normal follicles. Antrum formation was identified by the presence of a translucent cavity visible within the granulosa cell layers. The same operator performed all evaluations and measurements.

### 2.4. Assessment of Secondary Follicle Viability by Fluorescence Microscopy

After 18 days of culturing, the follicles (*n* = 20/treatment) were incubated in 100 μL drops of TCM-199 with 2 mM of ethidium homodimer-1 and 4 mM of calcein-AM (Molecular Probes, Invitrogen, Karlsruhe, Germany) for 15 min at 37 °C. After washing the follicles in TCM-199, they were analyzed under a fluorescence microscope (Nikon, Eclipse, TS 100, Yokohama, Japan). Follicles containing the cytoplasm of oocytes and granulosa cells positively labeled for calcein-acetoxymethyl ester (calcein-AM) (green) were classified as viable, while those with the chromatin stained with ethidium homodimer-1 (red) were not viable [21]. The intensity of fluorescence was evaluated using Image J software (National Institute of Health, Bethesda, MD, USA). Follicular staining intensity was calculated by measuring the pixels’ intensity in the area of the follicles after background subtraction.

### 2.5. Ultrastructural Analysis of Secondary Follicles

To evaluate the ultrastructure of the organelles and membranes in the oocyte and granulosa cells, transmission electron microscopy was performed in the 18-day-cultured secondary follicles. The follicles (*n* = 5 per treatment) were fixed in 2.5% glutaraldehyde and 4% paraformaldehyde in 0.1 M sodium cacodylate buffer (Karnovsky’s solution, pH 7.2) for 4 h at room temperature (~25 °C). Then, the follicles were embedded in 4% agarose and kept in sodium cacodylate buffer. The follicles were then post-fixed in 0.8% potassium ferricyanide, 1% osmium tetroxide, and 5 mM calcium chloride in 0.1 M sodium cacodylate buffer for 1 h at room temperature. After being washed in sodium cacodylate buffer, they were stained with 5% uranyl acetate. Dehydration was performed through a gradient of acetone solutions, and the follicles were embedded in epoxy resin (Epoxy Embedding Kit, Honeywell Fluka, Seelze, Germany). Then, semi-thin sections (2 μm) were cut, stained with toluidine blue, and evaluated using light microscopy. Then, ultrathin sections (70 nm) were counterstained with lead citrate and uranyl acetate and examined under a Morgagni FEI transmission electron microscope (Eindhoven, The Netherlands).

### 2.6. Statistical Analysis

The data on the follicular diameters, staining intensity of viability markers, and daily growth rate after 18 days of culturing were subjected to normal distribution analysis via the Kolmogorov–Smirnov test (GraphPad Prism software, version 9.0) (https://www.graphpad.com/scientific-software/prism/, accessed on 6 June 2022). Then, these data were compared by ANOVA and Kruskal–Wallis test, followed by Dunn’s test. Percentages of follicular growth, survival, and antrum formation were analyzed using Fisher’s exact test. Statistically significant differences were considered when *p* < 0.05.

## 3. Results

### 3.1. Effects of NAC on Follicle Growth, Morphology, and Antrum Formation

Table 1 shows that the presence of 1.0 mM NAC in the culture medium significantly increased the percentage of growing follicles after 18 days of culturing when compared to other treatments (*p* < 0.05). NAC, however, did not influence the percentage of morphologically normal follicles or antrum formation (*p* > 0.05). On the other hand, 25.0 mM NAC reduced the rate of growing follicles during culturing (*p* < 0.05).

Supplementation of the culture medium with different concentrations of NAC did not influence their growth (Table 2), but follicles cultured in all treatments had a significant increase in their diameters after 18 days.

### 3.2. Effects of NAC on Follicular Viability

Secondary follicles cultured in the presence of 1.0 mM NAC had higher fluorescence intensities for calcein-AM than those cultured in the control medium (*p* < 0.05) (Figure 1B). In contrast, follicles cultured in the presence of 5.0 and 25.0 mM NAC did not differ from each other or with those cultured in the control medium (*p* > 0.05). The fluorescence intensity for ethidium homodimer-1 was higher in follicles cultured in medium supplemented with 25.0 mM NAC when compared to those from the control group (*p* < 0.05) (Figure 1C). In addition, follicles cultured with of 25.0 mM NAC had stromal cells surrounding the follicles stained positively with ethidium homodimer-1 (Figure 1A).

### 3.3. Ultrastructural Characteristics of Secondary Follicles

Fresh secondary follicles (day 0) had oocytes with continuous oolemmae and microvilli attached to intact zonae pellucidae (Figure 2A). The ooplasm showed rounded mitochondria with peripheral cristae and regular mitochondrial membranes, as well as lipid droplets and endoplasmic reticula (Figure 2B). Granulosa cells from non-cultured follicles presented irregularly shaped nuclei and cytoplasm with developed endoplasmic reticula and a high number of elongated mitochondria with well-preserved cristae. A small number of lipid drops and vacuoles were also seen (Figure 2C).

The oocytes from follicles cultured in the control medium had intact zonae pellucidae in close association with oolemmae. The ooplasm, however, contained large vacuoles and mitochondria and endoplasmic reticula with signs of degeneration (Figure 3A). The ultrastructure of the granulosa cells of these follicles was well preserved and similar to those described for fresh follicles (Figure 3B).

The oocytes from follicles cultured in the presence of 1 mM NAC also had intact zonae pellucidae in close association with oolemmae, but the ooplasm showed a large number of vacuoles, vesicles, swollen endoplasmic reticula, and damage to the mitochondrial membranes and cristae. Multivesicular bodies and organelles with signs of degeneration were also seen (Figure 4A). The granulosa cells of these follicles were well-preserved, but the mitochondria had reduced numbers of cristae (Figure 4B).

Oocytes from follicles cultured with 5 mM NAC had broken zona pellucida and signs of granulosa-cell invasion. The organelles were generally no longer recognizable (Figure 5A). Granulosa cells were still well-preserved, but mitochondria had reduced numbers of cristae (Figure 5B).

In follicles cultured with 25 mM NAC, the oocyte cytoplasm was extremely vacuolated, with the vacuoles often fusing to produce a greater vacated area. The organelles were also unrecognizable (Figure 6A). Granulosa cells were detached from the zona pellucidae and from each other, and they had vacuoles and mitochondria with reduced numbers of cristae (Figure 6B).

## 4. Discussion

This study shows for the first time that 1 mM NAC increases calcein cytoplasmic staining and the percentage of growing secondary follicles after 18 days of culturing. Previous reports also showed that 1 mM NAC improves the quality of vitrified murine oocytes and the development of mouse embryos [22,23]. The thiol group of NAC provides electrons to free radicals [15,24], acting as a direct antioxidant. NAC is also a precursor of glutathione (GSH) by liberating a cysteine group and increasing the concentration of GSH in cells [15,24]. The GSH is a major component of intracellular antioxidant control that prevents injuries caused by oxidative stress in oocytes [25]. It is also interesting to note that several studies indicate that NAC promotes the survival and proliferation of various cell types (fibroblasts [26]; adipose-tissue-derived stem cells [27]; corneal endothelial cells [28]; intestinal epithelial cells [24]; and human retinal pigment epithelial cells [29]). Recently, Ding et al. [30] showed that NAC increases the production of transcripts for cyclin D1, which is related to cell proliferation, and this may have influenced the growth of the bovine secondary follicles in vitro. In contrast, the presence of 25 mM NAC reduced the percentages of growing follicles. Oocytes from follicles cultured with 5 or 25 mM NAC had poor ultrastructures and signs of the rupturing of zonae pellucidae. Sun et al. [31] showed that NAC in concentrations higher than 10 mM causes a reduction in the pH of the culture medium and is harmful to oocyte growth during in vitro maturation. Granulosa cells were, in general, well-preserved, but Ding et al. [30] demonstrated that NAC in concentrations higher than 10 mM suppressed porcine trophoblastic cell proliferation in vitro.

Regarding follicular viability, 1 mM NAC increased calcein cytoplasmic staining in cultured follicles. Calcein-AM is non-fluorescent and cell-membrane permeable, and it is converted to calcein (fluorescent form) when its AM ester group is cleaved by intracellular non-specific esterases [21,32]. Esterases are an important group of enzymes for animal cells, with a catalytic function in the reaction of ester hydrolysis, in particular, the acetylesterases that hydrolyze acetyl esters [33]. In the secondary and tertiary ovarian follicles of buffaloes, the activity of non-specific esterases in granulosa cells, corona radiata, and theca cells has been reported to be important for cell growth and viability [34]. Other studies have shown that NAC protects granulosa cells against apoptosis, as well as improving cell viability in human [17,19] and murine species [18]. The presence of 1 mM NAC in the culture medium also improves the viability of human chondrocytes [35]. The presence of 25 mM NAC in the culture medium, however, increases the percentages of cells with damaged membranes, as indicated by ethidium homodimer-1 staining. This marker also indicates an extravasation of cytoplasmic contents [36]. The oocytes from these follicles were extremely vacuolated, and the organelles were degenerated. The concentration of 25 mM NAC may be toxic for bovine secondary follicles; such an effect can be associated with a pro-oxidant action [37]. The same molecules that scavenge active free radicals can also cause oxidative stress, prompting the induction of cellular signaling to increase antioxidant defenses [38,39]. NAC acts as a pro-oxidant molecule when, for example, it comes into contact with transition metals and thus can react with H_2_O_2_ to generate a hydroxyl radical, which can cause lipid peroxidation and, consequently, cause damage to cell membranes [40,41]. Such a factor may have contributed to the reduction of the percentage of developing follicles after culturing in a medium supplemented with 25 mM NAC.

Ultrastructural analysis of cultured secondary follicles demonstrated that the oocyte is very sensitive to degeneration and that the culturing systems still need to be improved to maintain oocyte integrity. Sasaki et al. [42] demonstrated that oxidative stress in oocytes causes damage in many cellular components, such as in endoplasmic reticula, mitochondria, lipids, proteins, and enzymes. In goats, oocyte vacuolization and damage to mitochondria are the first signs of degeneration in preantral follicles after culturing [43]. Paulino et al. [5] also observed similar ultrastructural features in oocytes from bovine secondary follicles cultured in vitro for 18 days. The rupture of the zona pellucida and the invasion of granulosa cells into the oocyte may be due to prolonged days of in vitro culturing since the oocyte is the structure most susceptible to culturing conditions. The overexposure to oxygen, light, and temperature are involved in the increase of ROS levels, and in vitro environment variations consequently cause degeneration of oocytes [6,11,44]. During the in vitro culturing of secondary follicles, it has previously been demonstrated that degenerative signs appear initially in oocytes while granulosa cells continue to proliferate and remain healthy, emphasizing that the oocyte is far more sensitive to degeneration than are the granulosa cells [45,46]. Considering that only NAC was unable to improve oocyte ultrastructure, we cannot exclude the possibility that a combination of different antioxidants, with different mechanisms of action, is needed. Irrespective of treatment, granulosa cells did not show significant changes, only a reduction in the number of mitochondrial cristae. In preantral follicles, autophagy is involved in the degeneration of oocytes and later in the granulosa cells [47,48,49,50]. Autophagy is a process that promotes cellular homeostasis by which intracellular components, such as organelles and damaged proteins, are degraded and reused by the cell. However, it can cause autophagic cell death when there is a massive depletion of cytoplasmic contents [51]. According to Gannon et al. [52], autophagy can be triggered by oxidative stress and a loss of antioxidant capacity, and Furlong et al. [53] showed that oxidative stress activates adenosine monophosphate-activated protein kinase, resulting in autophagic cascade in the rat ovary.

## 5. Conclusions

In conclusion, 1 mM NAC increases both calcein cytoplasmic staining and the percentage of growing follicles. The presence of 5 or 25 mM NAC, however, causes damage in the cellular membranes and organelles. A concentration of 25 mM NAC reduces the percentage of growing follicles.

## Figures and Tables

**Figure 1 animals-12-03190-f001:**
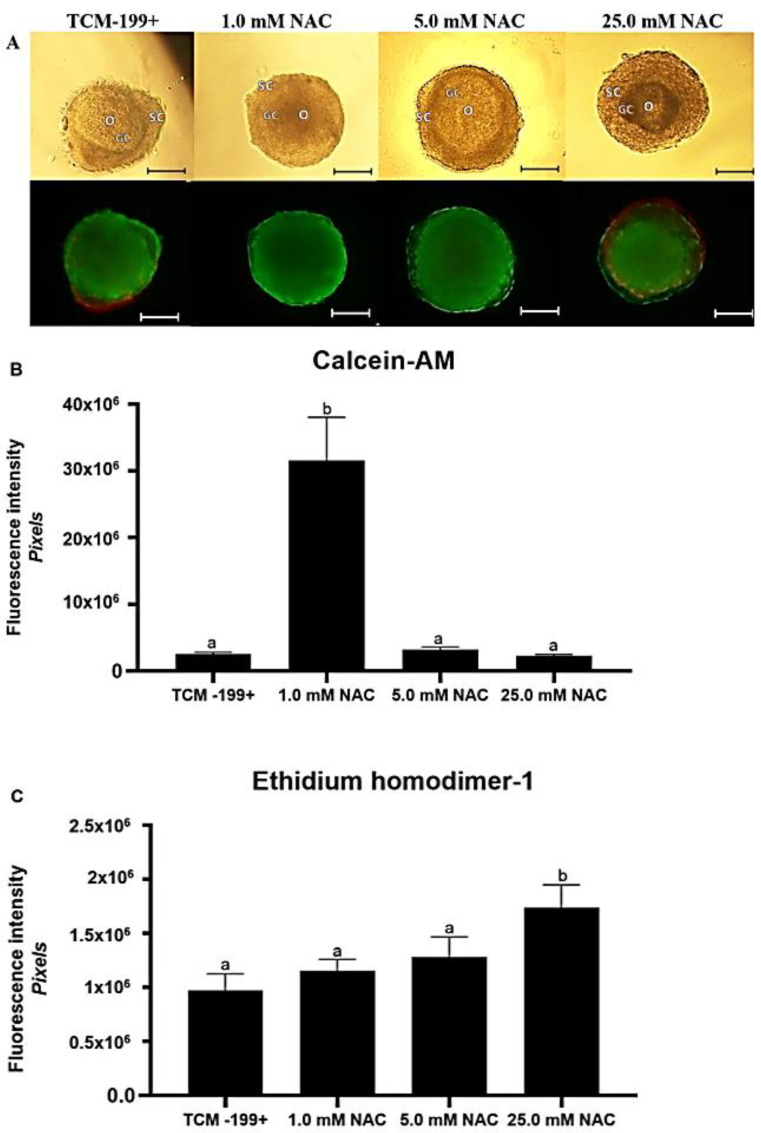
Morphology of secondary follicles after culturing in the different treatments (**A**). Staining intensity (mean ± SEM) for calcein-AM (**B**) and ethidium homodimer-1 (**C**) in bovine secondary follicles after 18 days of culturing in TCM-199^+^ alone or with different concentrations of NAC (1.0, 5.0, and 25.0 mM). Lower case letter “a, b” represent significant differences between treatments (*p* < 0.05). O: oocyte, SC: stromal cells, GC: granulosa cells. Scale bar: 100 µm.

**Figure 2 animals-12-03190-f002:**
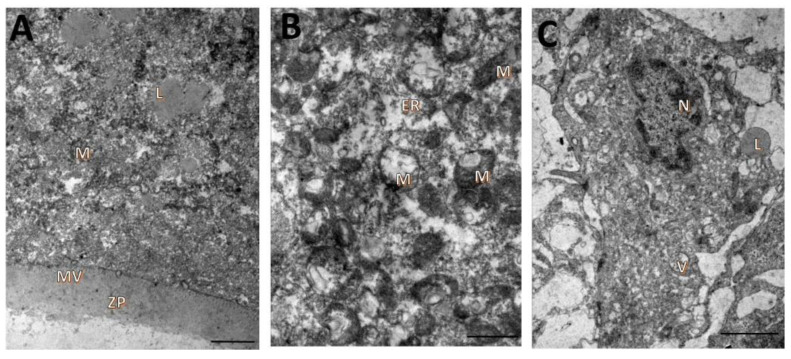
Ultrastructural characteristics of oocytes (**A**,**B**) and granulosa cells (**C**) from uncultured secondary follicles. Mitochondria (M), zona pellucida (ZP), nucleus (N), lipid droplets (L), vacuoles (V), microvillous (MV), endoplasmic reticulum (ER). Scale bar = 100 µm.

**Figure 3 animals-12-03190-f003:**
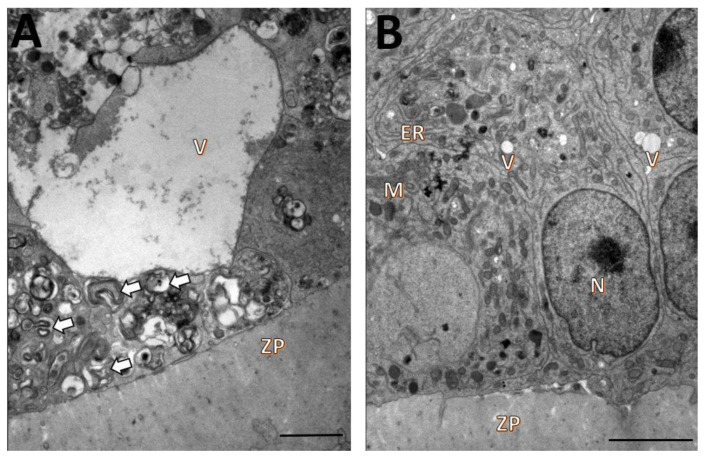
Ultrastructural characteristics of oocytes (**A**) and granulosa cells (**B**) from secondary follicles after 18 days of in vitro culturing in TCM 199^+^ alone. Mitochondria (M), zona pellucida (ZP), nucleus (N), vacuoles (V), endoplasmic reticulum (ER), degenerating organelles (arrows). Scale bar: 100 µm.

**Figure 4 animals-12-03190-f004:**
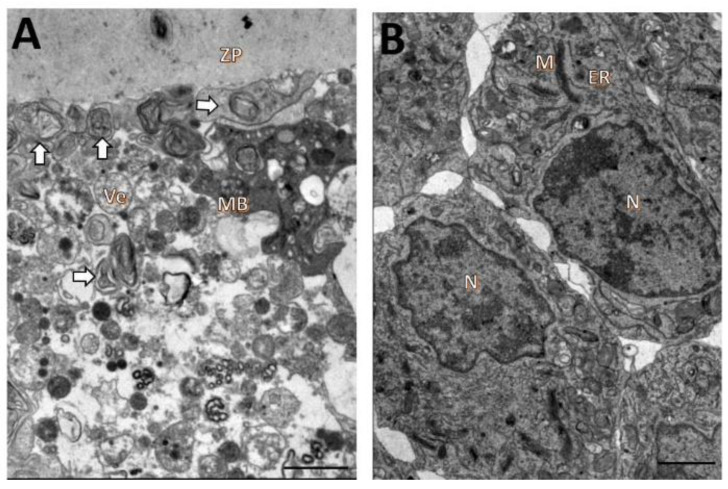
Ultrastructural characteristics of oocytes (**A**) and granulosa cells (**B**) from secondary follicles after 18 days of in vitro culturing in TCM 199^+^ supplemented with 1 mM NAC. Mitochondria (M), zona pellucida (ZP), nucleus (N), endoplasmic reticulum (ER), vesicles (Ve), multivesicular body (MB), degenerating organelles (arrows). Scale bar: 100 µm.

**Figure 5 animals-12-03190-f005:**
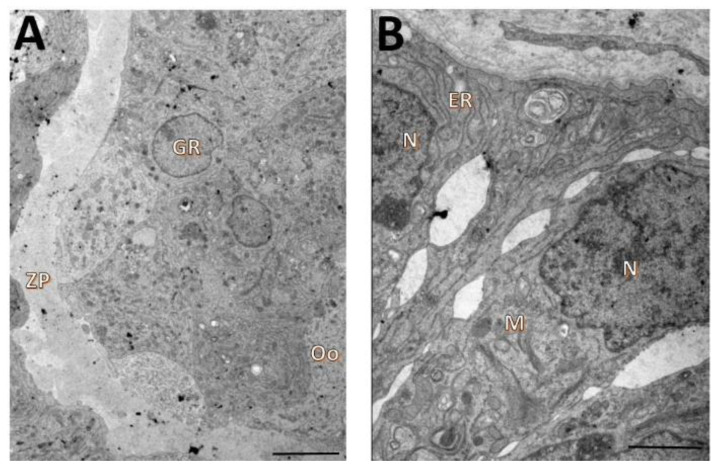
Ultrastructural characteristics of oocytes (**A**) and granulosa cells (**B**) from secondary follicles after 18 days of in vitro culturing in TCM 199^+^ supplemented with 5 mM NAC. Ooplasm (Oo), mitochondria (M), zona pellucida (ZP), nucleus (N), endoplasmic reticulum (ER), granulosa cells (GR). Scale bar: 100 µm.

**Figure 6 animals-12-03190-f006:**
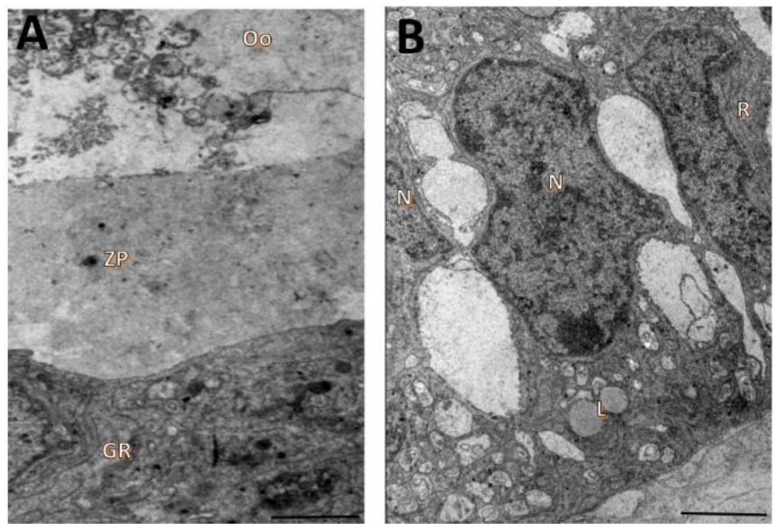
Ultrastructural characteristics of oocytes (**A**) and granulosa cells (**B**) from secondary follicles after 18 days of in vitro culturing in TCM 199^+^ supplemented with 25 mM NAC. Ooplasm (Oo), zona pellucida (ZP), nucleus (N), lipid droplets (L), granulosa cells (GR). Scale bar: 100 µm.

**Table 1 animals-12-03190-t001:** Percentages of morphologically normal follicles, antrum formation, and growing follicles after 18 days of in vitro culturing in TCM-199^+^ alone or with different concentrations of NAC.

Treatments	Morphologically Normal Follicles	Antrum Formation	Growing Follicles
TCM-199^+^	97.97% (97/99)	20.20% (20/99)	62.63% (62/99) a
NAC 1.0 mM	100.00% (101/101)	29.70% (30/101)	76.24% (77/101) b
NAC 5.0 mM	100.00% (99/99)	24.24% (24/99)	54.55% (54/99) ac
NAC 25.0 mM	98.98% (98/99)	29.29% (29/99)	47.47% (47/99) c

a, b, c: Lowercase letters represent significant differences between Day 0 and Day 18 (*p* < 0.05).

**Table 2 animals-12-03190-t002:** Diameters (mean ± SEM) of bovine secondary follicles after 0 and 18 days of in vitro culturing in TCM-199^+^ alone or supplemented with different concentrations of NAC.

Treatments	Day 0	Day 18	Growth Day 0–18	Daily Growth Average (μm)
TCM-199^+^	203.69 ± 3.52 a (*n* = 62)	279.46 ± 8.21 b (*n* = 62)	75.77 ± 7.46	4.21 ± 0.41
NAC 1.0 mM	201.76 ± 3.04 a (*n* = 77)	280.77 ± 8.76 b (*n* = 77)	82.80 ± 8.51	4.39 ± 0.43
NAC 5.0 mM	203.40 ± 3.41 a (*n* = 54)	265.92 ± 8.17 b (*n* = 54)	66.42± 8.41	3.46 ± 0.41
NAC 25.0 mM	210.95 ± 4.15 a (*n* = 47)	271.18 ± 8.81 b (*n* = 47)	60.24 ± 7.63	3.34 ± 0.42

a, b: Lowercase letters represent significant differences between days 0 and 18 (*p* < 0.05).

## Data Availability

The data presented in this study are openly available in the repository Federal University of Ceará (https://repositorio.ufc.br/).

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
