# Peer review of "Effects of N-acetylcysteine on Growth, Viability, and Ultrastructure of In Vitro Cultured Bovine Secondary Follicles"

_animals, 2022, doi:10.3390/ani12223190_

Round 1

Reviewer 1 Report

L 17-18: what does the sentence mean?

L66: what is the cattle breeds, age and the condition of ovary?

L75: this sentence is not intact? How to isolate the preantral follicles from the ovary?

L83: what is the volume of microdrops? What is the percentage of the 60 μL culture medium in the microdrops?

L102:  how many replicates of the ovary n=60 (L66) or the follicle n=20 in this study?

L113-114: what does the “transmission electron microscopy was performed” mean? And “the isolated follicles” should be “the cultured follicles”?

L160: use different color of arrow to indicate the cavity, ZP or oocyte on the picture in Fig1 A.

L166-211: the pictures from the transmission electron microscopy should be put together to compare the differences of follicular ultrastructure.

L217,L 237L287: calcein staining indicated that positive cytoplasm, here ,it should be delete “calcein staining and”.

Author Response

Reviewer 1

L 17-18: what does the sentence mean?

Answer: The sentence was adjusted to clarify the meaning.  “On the other hand, follicles cultured with 25.0 mM NAC had higher fluorescence intensity for ethidium homodimer-1, which is a sign of degeneration.” (L. 31-33)

L66: what is the cattle breeds, age and the condition of ovary?

Answer: The sentence “Bovine (Bos taurus) ovaries (n=60) were collected in a local slaughterhouse (Sobral, Ceará, Brazil)…” was replaced by “Bovine (Bos taurus) ovaries (n=60) containing visible antral follicles were collected in a local slaughterhouse (Sobral, Ceará, Brazil) from mixed breed cows (age between 4 - 12 years).” (L.81-83)

L75: this sentence is not intact? How to isolate the preantral follicles from the ovary?

Answer: We have adjusted the sentence. “For follicular isolation, the methodology of Vasconcelos et al. [8] was followed. Fragments (1-2 mm) of the ovarian cortex were removed with the aid of a sterile scalpel blade and placed in TCM-199 medium supplemented with HEPES (0.05 mM / mL) and penicillin (100IU) / streptomycin (0.1 mg/mL). Subsequently, the fragments were analyzed in a stereomicroscope (SMZ 645 Nikon, Tokyo, Japan) to visualize the secondary follicles (150-250 μm).  (L.92-97)

L83: what is the volume of microdrops? What is the percentage of the 60 μL culture medium in the microdrops?

Answer: 100 μL. This information was added at L. 100

L102:  how many replicates of the ovary n=60 (L66) or the follicle n=20 in this study?

Answer:

The sentence: “This experiment was repeated eight times.” Was replaced by “This experiment was repeated eight times and a total of 398 follicles were isolated and cultured.” (L. 109-110)

L113-114: what does the “transmission electron microscopy was performed” mean? And “the isolated follicles” should be “the cultured follicles”?

Answer: It was performed to evaluate the ultrastructure of membranes and organelles in oocyte and granulosa cells (L.131). “Isolated follicles” was replaced by “cultured follicles” (L.133)

L160: use different color of arrow to indicate the cavity, ZP or oocyte on the picture in Fig1 A.

Answer: The suggestion was accepted. Oocyte, granulosa cells and stromal cells are now identified in this figure (L. 183).

L166-211: the pictures from the transmission electron microscopy should be put together to compare the differences of follicular ultrastructure.

Answer: Dear reviewer, we tried to put the figures together. However, putting 11 images together in a single page make each image very small, which difficult identification of organelles. Thus, we decided not to change them.

L217, L237 L287: calcein staining indicated that positive cytoplasm, here, it should be delete “calcein staining and”.

Answer: Revised at Lines 235, 255 and 305.

Reviewer 2 Report

In this manuscript, the effect of N-acetylcysteine on growth, viability and ultrastructure of in vitro cultured bovine secondary follicles was evaluated. The experimental design is sound and results contribute to increase the knowledge in the field.THe manuscript is clear, discussions are appropriate and conlusions are relevant to the results. Only minor revisions are required.

Introduction:

Line 46 and 57: It has been reported

M&M:

Lack of provenience of media and reagents.

Line 75: the methodology of Vasconcelos et al. [8]

Line 76: sterile scalpel blade

Line 89: 5% CO2 in air

Line 90: were replaced

Line 103: 100 μl drops

Line 107: use full name of calcein AM at first appearance

Discussion:

Line 227: [30] have

Line 247; also improves

Line 250: delete spaces

Author Response

Response to Reviewer 2

Introduction:

Line 46 and 57: It has been reported

Answer: Done (Lines 60 and 71).

M&M:

Lack of provenience of media and reagents.

Answer: We added this information at L. 80-81.

Line 75: the methodology of Vasconcelos et al. [8]

Answer: Done (L. 92) Thanks for the suggestion

Line 76: sterile scalpel blade

Answer: Revised (L.93)

Line 89: 5% CO2 in air

Answer: Done (L. 106)

Line 90: were replaced

Answer: Done (L. 107)

Line 103: 100 μl drops

Answer: Done (L. 121)

Line 107: use full name of calcein AM at first appearance

Answer: Done (L. 125)

Discussion:

Line 227: [30] have

Answer: Done (L. 246)

Line 247; also improves

Answer: Revised (line 265)

Line 250: delete spaces

Answer: Revised
